# Oral Undifferentiated Pleomorphic Sarcoma: A Novel *SPECC1L::TERT* Gene Fusion and a Comprehensive Literature Review

**DOI:** 10.3390/genes16070830

**Published:** 2025-07-17

**Authors:** Mario Della Mura, Joana Sorino, Eugenio Maiorano, Gerardo Cazzato, Anna Colagrande, Alfonso Manfuso, Concetta Caporusso, Chiara Copelli, Eliano Cascardi

**Affiliations:** 1Section of Molecular Pathology, Department of Precision and Regenerative Medicine and Ionian Area (DiMePRe-J), University of Bari “Aldo Moro”, 70124 Bari, Italy; mariodellamura1@gmail.com (M.D.M.); j.sorino@studenti.uniba.it (J.S.); eugenio.maiorano958@gmail.com (E.M.); gerardo.cazzato@uniba.it (G.C.); concetta.caporusso@policlinico.ba.it (C.C.); eliano20@hotmail.it (E.C.); 2Maxillofacial Surgery Unit, Interdisciplinary Department of Medicine, University of Bari “Aldo Moro”, 70124 Bari, Italy; a.manfuso@operapadrepio.it (A.M.); chiara.copelli@uniba.it (C.C.)

**Keywords:** undifferentiated pleomorphic sarcoma, malignant fibrous histiocytoma, sarcoma, oral cavity, maxilla

## Abstract

**Background:** Undifferentiated pleomorphic sarcoma (UPS) is a rare, high-grade soft-tissue sarcoma characterized by a patternless proliferation of bizarre pleomorphic tumor cells lacking identifiable lineage differentiation. Its occurrence in the oral cavity is exceptionally uncommon and poses significant diagnostic challenges due to its morphological overlap with a wide spectrum of other malignancies. **Material and Methods**: We report a novel case of oral UPS in a 54-year-old woman, characterized by an exceptionally large size and a rapidly progressive clinical course. The diagnostic evaluation included clinical, radiological, histopathological, immunohistochemical, and molecular analyses conducted within a multidisciplinary framework. A comprehensive review of the literature on oral UPS was also performed. **Results**: The patient underwent an aggressive demolitive surgical approach due to the extent of the lesion. Molecular analysis revealed a previously unreported *SPECC1L::TERT* gene fusion. The literature review highlighted the rarity of oral UPS, its geographic predilection for Central and East Asia, possible associations with traumatic events, and its heterogeneous clinical and histopathological presentations. **Conclusions**: This case underscores the critical importance of a thorough diagnostic workup to ensure the accurate diagnosis and appropriate management of this rare and aggressive tumor. Multidisciplinary evaluation is essential, especially in anatomically complex and diagnostically challenging presentations such as oral UPS.

## 1. Introduction

Undifferentiated pleomorphic sarcoma (UPS), formerly known as malignant fibrous histiocytoma (MFH), is a type of undifferentiated soft-tissue sarcoma consisting of a patternless growth of pleomorphic bizarre tumor cells showing no identifiable line of differentiation. The earliest reports suggested it was the most common soft-tissue sarcoma in the adult population. However, subsequent analyses incorporating immunohistochemistry, cytogenetics, and molecular biology have demonstrated that many of those malignancies are better classified as other types of sarcomas. Currently, it represents a diagnosis of exclusion after the possibility of other dedifferentiated mesenchymal and non-mesenchymal neoplasms has been ruled out [1,2].

UPS is anatomically ubiquitous, originating from either soft tissue or bone, with the most frequent locations represented by the extremities and retroperitoneum [1]. It can very rarely involve the oral cavity, constituting a significant challenge from both the diagnostic and therapeutic points of view. Indeed, the latter may be affected by a wide and heterogeneous spectrum of malignancies arising from different cell lineages [3,4,5,6]; therefore, an accurate pathological workup including extensive sampling and careful histological and immunohistochemical evaluation is required before the diagnosis can be rendered. Moreover, complete surgical resection with clear margins is the current standard of therapy; however, it is often not easy to achieve due to the size and anatomical location of neoplasms, as well as the surgical complexity of the oral–maxillofacial region. Finally, well-established recommendations about the need for adjuvant therapy do not exist because of the rarity of the condition and the consequent lack of multicentric studies.

Herein, we report a novel case of UPS of the oral cavity, characterized by an exceptionally large size and aggressive clinical course, with the aim of expanding the current understanding about this entity. Moreover, we provide a comprehensive review of the literature in order to deepen the etiopathogenesis, pathological features, and clinical-prognostic correlation of UPS of the oral cavity.

## 2. Materials and Methods

The patient’s informed consent was obtained. The clinical history was retrieved from medical records. Tissue samples were formalin-fixed, paraffin-embedded (FFPE), and stained with Hematoxylin and Eosin (H&E). Immunohistochemical staining was performed using antibodies against cytokeratin AE1/AE3 (CK AE1/AE3), MNF116 (CK MNF116), epithelial membrane antigen (EMA), p63, p40, CD10, CD163, PAX8, CD34, ERG, Melan-A, S-100, vimentin, smooth muscle actin (SMA), smooth muscle myosin (SMMS-1), desmin, and Ki-67 to further characterize the neoplastic cells. Fluorescence in situ hybridization (FISH) analysis to detect *MDM2* gene amplification was also performed.

RNA-based Next-Generation Sequencing (NGS) was conducted using the Archer FusionPlex OPBG Custom V2 panel to identify gene fusions and hotspot mutations.

A literature review was performed using the PubMed, Scopus, and Web of Science databases. The search strategy included the following keywords: undifferentiated pleomorphic sarcoma, malignant fibrous histiocytoma, oral cavity, mouth, maxilla, palate, mandible, bucca, cheek, tongue, and gingiva. Only English-language articles published from 2002 onward—corresponding to the major revision of the UPS category in the WHO classification of soft tissue and bone tumors—were considered. Studies were excluded if they involved non-oral sites, lacked essential clinical or histopathological data, or described tumors now reclassified as other entities (e.g., myxoid MFH, currently classified as myxofibrosarcoma).

## 3. Case Report

A 54-year-old Caucasian woman was admitted to our institution with a large swelling of the left cheek region (Figure 1A). She reported symptom onset approximately four months earlier, initially managed as a presumed abscess with antibiotic therapy. The patient denied any history of trauma. Due to a lack of clinical improvement, a biopsy was performed at another center, which yielded a diagnosis of undifferentiated mesenchymal neoplasm with a giant cell component, high proliferative activity, and an “only vimentin-positive” phenotype.

As a result, the patient underwent two cycles of chemotherapy with epirubicin and ifosfamide; however, the lesion continued to enlarge. Upon presentation to our institution, intraoral examination revealed restricted mouth opening and a bulky 10 cm mass occupying the left cheek and masticatory space. Further imaging studies, including MRI, CT, and whole-body PET-CT, were performed. The contrast-enhanced CT (Figure 1B) revealed a large, heterogeneous, expansile mass occupying the left maxilla and protruding into the oral cavity, with invasion of the maxillary sinus, ipsilateral nasal cavity, and lower extracompartmental extension on the external side (the lesion reached the subcutaneous tissue without infiltrating it). No evidence of mandibular bone involvement was identified. The PET-CT ruled out both regional and distant metastases.

After multidisciplinary team discussion, surgical intervention was indicated. The patient underwent a Weber–Ferguson approach combined with osteotomy of the zygomatic arch to access the pterygoid fossa, followed by a left radical maxillectomy including resection of the ipsilateral masseter and pterygoid muscles, ethmoid bone, and left orbital floor. Reconstruction was achieved using a titanium mesh for the orbital floor and a myocutaneous anterolateral thigh free flap. The postoperative course was uneventful, and the patient was discharged after 14 days of hospitalization.

Histopathological examination of the resection specimen revealed a highly cellular, extensively ulcerated neoplasm, diffusely infiltrating the maxilla to the bone. The tumor consisted of pleomorphic, bizarre, atypical cells with markedly irregular nuclei, with vesicular chromatin, prominent nucleoli, and eosinophilic-to-amphophilic cytoplasm; scattered multinucleated giant cells were also noted. The neoplastic cells were arranged in a vaguely storiform pattern within a collagenous stroma containing numerous thin-walled ectatic vessels and histiocytes (Figure 2). The mitotic activity was brisk (up to 22 mitoses/mm^2^ in hot-spot areas), including atypical forms. Extensive necrosis was also observed. No evidence of lymphatic, vascular, or perineural invasion was identified.

The immunohistochemical analysis failed to demonstrate specific lineage differentiation. The tumor cells were negative for CK AE1/AE3, CK MNF116, EMA, p63, p40, PAX8, CD34, ERG, Melan-A, S-100, SMA, SMMS-1, and desmin. Vimentin was diffusely positive. CD10 showed weak positivity in the scattered tumor cells, while CD163 marked background stromal histiocytes. The Ki-67 proliferation index was approximately 80%. The FISH revealed no *MDM2* gene amplification.

The molecular analysis detected a novel gene fusion transcript: *SPECC1L::TERT*.

The patient subsequently underwent to adjuvant radiotherapy due to positive resection margins. As of 8 months, she is alive and disease-free.

## 4. Discussion

UPS is a type of undifferentiated soft-tissue sarcoma that very rarely involves the oral cavity, characterized by a non-specific clinical presentation, uniquely exclusion-based histopathological definition, uncertain histogenesis, and an aggressive clinical course.

A total number of 54 oral UPSs have been reported in the English literature so far, including our case (Table 1). Affected patients are more frequently males (35 males vs. 19 females, M:F = 1.8:1), with a mean age of 54 years (range: 8–88) [7,8]. A striking geographic predilection for middle-east Asia emerges, with the majority of cases reported in India (18/54, 33.3%), followed by China (10/54, 18.5%), Japan (6/54, 11.1%), and Iran (3/54, 5.6%) (Figure 3). UPS can originate from both soft tissue and bone; the most common site of presentation within the oral cavity is the mandible (22/54, 40.7%), followed by the maxilla (14/54, 25.9%).

Clinically, they usually present as rapidly growing masses, frequently ulcerated, without distinctive macroscopic features, ranging in size from 2 [31] to 10 cm (our case), with an average of 4.9 cm.

The radiographic features are non-specific and can vary from a predominant radiolucency to mixed radiolucent/radio-opaque appearance, depending on the degree of bone destruction and the presence of calcification or ossification [34]; pathological fractures can coexist [8]. In a study conducted on 13 patients, Park et al. reported the main characteristics of UPS of the head and neck on tomographic imaging. In particular, in CT scans, it appeared as a lobulated soft tissue mass, iso-attenuated to muscle, that could display either heterogeneous or homogeneous enhancement, sometimes with central attenuation due to necrosis or hemorrhage; in 5–20% calcifications were also found. In MRI, it was isointense to muscle on T1 weighted images and heterogeneously hyperintense on T2 weighted images [44].

Due to the lack of distinctive clinical and imaging features, pathological examination is the gold standard for definitive diagnosis and can be rendered only in presence of a proliferation of highly atypical pleomorphic cells in a storiform, fascicular, or patternless arrangement, showing no morphological or immunohistochemical line of differentiation [2]. Neoplastic cells usually show positivity toward vimentin; variable reactivity to histiocytic markers as CD68, alpha-1-antichimotrypsin (1ACT), alpha-1-antitrypsin (A1AT), and CD163; and negativity toward other lineage specific antigens. Occasionally, focal reactivity to smooth muscle markers can be observed; however, such findings should not be interpreted as true myogenic differentiation, which is further supported by negativity to h-caldesmon. Moreover, focal aberrant expression of cytokeratins has been also rarely reported [8,9,13,19,39].

UPS may provide difficulties in the differential diagnosis of other mesenchymal and non-mesenchymal neoplasms that are mandatory to exclude.

Between mesenchymal neoplasms, fibrosarcoma, pleomorphic leiomyosarcoma, pleomorphic rhabdomyosarcoma, pleomorphic liposarcoma, and dedifferentiated liposarcoma must be ruled out. Fibrosarcoma may contain areas with a storiform pattern, but it typically lacks significant pleomorphism as well as bizarre multinucleated giant cells. Pleomorphic leiomyosarcoma and rhabdomyosarcoma can be separated from UPS by the presence of deeply eosinophilic cytoplasm, longitudinal myofibrils, or cross-striations and/or diffuse immunoreactivity with multiple myogenic markers. Pleomorphic liposarcoma may contain large numbers of pleomorphic giant cells; however, the characteristic finding is the presence of multivacuolated pleomorphic lipoblasts, possibly showing CD34 and S100 positivity. Dedifferentiated liposarcoma arises from a well-differentiated component, which is seen almost focally at the edge of the lesion, and carries *MDM2* gene amplification.

On the other hand, non-mesenchymal neoplasms that morphologically overlap with UPS are essentially sarcomatoid carcinoma, melanoma, and anaplastic lymphoma. To the first aim, a battery of epithelial markers including broad-spectrum cytokeratins is required, but equivocal results can occur. First, in sarcomatoid carcinomas, the immunohistochemical expression of epithelial markers can be only focal. Second, virtually any type of sarcoma, including UPS, can, on occasion, express cytokeratins. In such cases, strong and diffuse cytokeratin staining (especially with multiple antibodies), the expression of other epithelial markers such as EMA or p63, and/or the recognition of an intraepithelial dysplastic component will strongly support the diagnosis of sarcomatoid carcinoma. Furthermore, screening for almost two melanocytic markers (such as HMB45 and Melan A) as well as CD45/LCA or CD30 helps to exclude a melanocytic or hematolymphoid origin [2,23,43].

The etiology of UPS is not well-understood to date. Genetic background and environmental factors such as a previous history of trauma or radiotherapy, as well as malignant transformation from benign lesions, have been proposed. In our review, two patients were affected by neurofibromatosis type 1 (NF1) syndrome (2/54, 3.7%) [12,14]. A history of antecedent trauma has been reported in 11 cases (11/54, 20.4%), mostly consisting of dental extractions or previous fractures, suggesting that some of these tumors may originate from an initial proliferative response to a traumatic injury. In fact, following a *noxa*, fibrocytes and other adult stem cells arrive at the site of tissue damage and transdifferentiate into various other elements; it is possible that the stimulation of these cells in the wound milieu, followed by cycles of repair and damage, can occasionally lead to their malignant transformation [16]. Finally, five cases of radiation-induced (5/54, 9.3%) oral UPS have been reported [20,29,39,40], as well as one case arising from a previous benign fibrous histiocytoma (1/54, 1.9%) [18].

There are currently two main hypotheses regarding the histogenesis of UPS, not mutually exclusive to each other. The first postulates that it does not actually represent a real biological entity but rather a common “morphologic pattern” shared by different mesenchymal malignancies that can be actually unrelated but share similar morphological features as the result of a final common pathway of cancer progression; in this model, it was presumed that the tumors would become progressively more undifferentiated, ultimately resulting in high-grade undifferentiated pleomorphic sarcoma, as happens in poorly differentiated carcinomas as well. The second theory states that UPS is not the result of loss of differentiation from a previously differentiated sarcoma but derives from the neoplastic transformation of very immature mesenchymal stem cells, as happens in embryonal carcinoma or leukemia; according to this model, transformation at the early points of connective tissue differentiation would result in less-differentiated sarcomas (as happens in UPS), while transformation at later time points of differentiation would result in well-differentiated sarcomas [1].

The molecular landscape of the neoplasm is also quite heterogeneous, with no characteristic mutations identified. In our case, a novel gene fusion transcript has been identified: *SPECC1L::TERT*. This molecular alteration has not been reported in the literature so far, and its role in oncogenesis remains to be elucidated.

Sperm antigen with calponin homology and coiled-coil domain 1-like (SPECC1L) is a critical cytoskeletal scaffolding protein involved in actin cytoskeleton reorganization during morphogenesis. Although SPECC1L is not essential for the initiation or completion of tissue movement and fusion events, it appears to modulate the efficiency and dynamics of these processes. It contains several coiled-coil domains, as well as binding sites for key cytoskeletal proteins such as F-actin, microtubules, and non-muscle myosin II (NMII). Intriguingly, germline mutations in SPECC1L are associated with craniofacial abnormalities, including oblique facial clefts, cleft palates, and severe bilateral ocular hypoplasia, while its role in tumorigenesis remains poorly understood [46].

The human ribonucleoprotein telomerase complex consists of a core catalytic subunit, telomerase reverse transcriptase (TERT), and an independently encoded RNA component (TERC or TR), which, together with a set of associated proteins, enables the extension of telomeric DNA. Telomerase is normally active during embryonic development in stem cells but becomes silenced upon differentiation in somatic cells, thereby limiting cellular proliferative capacity through progressive telomere shortening. This telomere erosion-induced senescence serves as a critical tumor-suppressive barrier. In contrast, telomerase reactivation—primarily through increased expression of TERT—is observed in the vast majority of human cancers, representing a common mechanism for cellular immortalization [47].

To date, the functional implications of the SPECC1L::TERT fusion transcript remain unknown. No data regarding this alteration are available in the medical literature or in genomic databases (e.g., OncoKB), nor have in silico models been developed. We hypothesize that this fusion may lead to upregulated TERT expression and/or contribute to cytoskeletal remodeling, thereby enhancing tumor cell motility and increasing invasive potential. However, our panel was not suitable for identifying the exact protein domains involved in the fusion, and thus we were unable to determine its precise molecular mechanism. In summary, the biological relevance and oncogenic potential of this novel molecular alteration remain largely unexplored and warrant further investigation.

The biological behavior of UPS of the oral cavity is not completely understood due to its rarity. They are morphologically high-grade lesions and are characterized by an aggressive clinical course.

In general, the treatment of affected patients involves complete surgical removal of the neoplasm with microscopically negative margins. It represents definitive treatment in the great majority of our reviewed cases (47/54, 87.0%). Since regional lymph node metastases are decidedly uncommon (2/54, 3.7% in our series) [17], neck dissections are not advocated in most cases. Radiotherapy is preferentially recommended in the presence of positive or close margins (<1 cm), bone, major vessels, or nerve involvement, as well as unresectable disease. The role of chemotherapy is mostly limited to non-surgical and/or metastatic disease, while in all other cases, the opportunity for pre- or postoperative antineoplastic treatment is questionable and should be subjected to multidisciplinary evaluation [2]. In our case, neoadjuvant chemotherapy with epirubicin and ifosfamide was administered in order to reduce the tumor size and allow for a less demolitive surgery. However, it proved ineffective, as the tumor continued to grow during treatment, confirming that upfront surgery remains the preferred approach whenever feasible. Moreover, we also decided to perform postoperative radiotherapy due to the positive resection margins.

Follow-up was available in 42 of the 54 patients in our series, with an average follow-up time of 18.9 months (range: 2–96 months). Of them, 18 died of disease (18/42, 42.8%) and 5 experienced recurrences after initial therapy (5/42, 11.9%), occurring within 2 years from the first diagnosis (average: 12.8 months); only 15 patients were alive and disease-free (15/42, 35.7%). The most common metastatic site was the lung (6/54, 11.1%), while other locations (brain, mediastinum, liver, bone, stomach, pancreas) have been only rarely reported.

## Figures and Tables

**Figure 1 genes-16-00830-f001:**
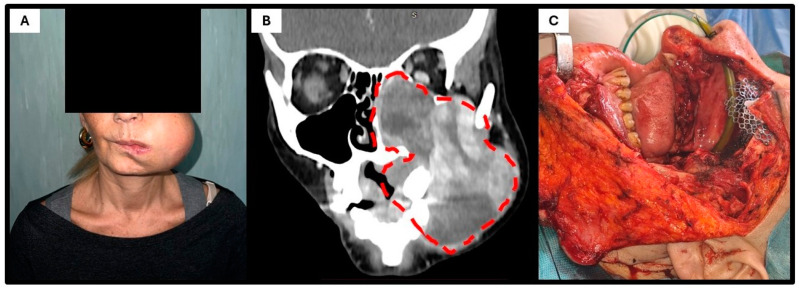
Clinical–radiological presentation of the patient. (**A**) Clinical photograph at admission showing a bulky mass occupying the left cheek. (**B**) Contrast-enhanced CT scan demonstrating a large, heterogeneous soft-tissue mass eroding the left maxilla, with invasion of the maxillary sinus, ipsilateral nasal cavity, and orbital floor and extracompartmental extension into the subcutaneous tissue (dot lines). (**C**) Intraoperative image following left radical maxillectomy, showing the surgical exposure of the tumor bed prior to reconstruction.

**Figure 2 genes-16-00830-f002:**
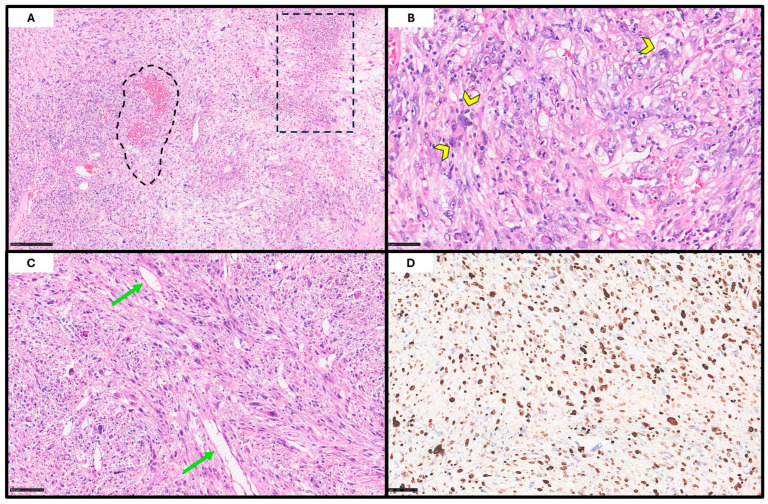
Histopathological features. At low-power magnification ((**A**); H&E; scale bar, 250 μm), a highly cellular neoplasm with extensive necrotic and hemorrhagic areas can be observed (dot lines). At higher magnification ((**B**,**C**); H&E; scale bars, 50 and 100 μm), the tumor is composed of pleomorphic, atypical cells arranged in a vaguely storiform pattern, embedded within a scant stroma rich in ectatic, thin-walled vessels (arrow); scattered multinucleated giant cells (arrowhead) are also present. Immunohistochemical staining for Ki-67 ((**D**); scale bar, 50 μm) demonstrates a high proliferative index (~80%), consistent with the aggressive nature of the lesion.

**Figure 3 genes-16-00830-f003:**
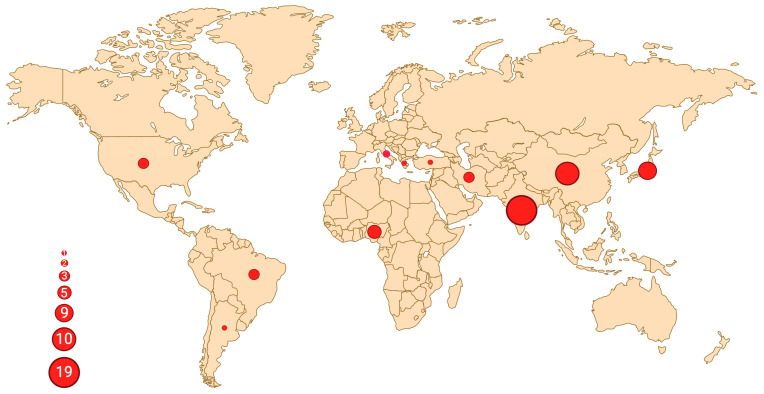
Geographic distribution of reported oral UPS cases. Each dot represents a country with at least one published case, and the size of the circle is proportional to the number of reported cases. The highest concentration can be observed in India (n = 19), followed by China (n = 10) and Japan (n = 6), with a total of 38 cases (38/54, 70%) reported in Asia. Created in BioRender. Crimini, E. 2025 https://BioRender.com/tiq8w31, accessed on 12 June 2025.

**Table 1 genes-16-00830-t001:** This table summarizes the 54 reported cases of oral UPS from 2002 to date, in chronological order, including our case. The reports are listed in chronological order and include information on geographic locations, histories of trauma, treatment, and follow-up. Legend: aCT: adjuvant chemotherapy; aRT: adjuvant radiotherapy; DF: disease-free; DOD: died of disease; nANLO: anlotinib as neoadjuvant therapy; nCT: neoadjuvant chemotherapy; NR: not reported; PR: partial response; R: recurrence; S: surgery.

No	Paper ID	Country	Age	Sex	Site	History of Trauma	Tumor Size (cm)	Metastasis	Treatment	Follow-Up (Months)
1	[9]	Greece	24	M	Left tongue	NR	3	No	S	18 (DF)
2	[10]	Turkey	32	F	Left mandible	No	5	NR	S	2 (R)
3	[11]	Brazil	56	M	Right posterior mandible	NR	4.5	No	S + aRT	25 (DF)
4	[12]	Japan	61	M	Left bucca	NR	5	No	RT	22 (PR)
5	[13]	Japan	68	M	Left posterior mandible	NR	NR	Lung, liver	S	8 (DOD)
6	[14]	Brazil	72	F	Left posterior mandible	NR	5	Femur	S	NR
7	[15]	India	20	F	Labial superior gingiva	NR	2.5	NR	S	3
8	[16]	Iran	38	M	Mandibular symphysis	Yes	NR	Brain	S + aRT	12 (R)
9	[17]	USA	37	F	Anterior mandible	No	9.5	Laterocervical lymphnodes, lung	S	12 (DOD)
10	[18]	India	72	M	Labial superior mucosa	Yes	9	NR	S	NR
11	[19]	Japan	80	M	Right mandible	Yes	6	No	None	autopsy diagnosis (DOD)
12	[20]	Japan	79	M	Left maxilla	No	5.5	NR	S + aCT	19 (DOD)
13	[21]	India	66	M	Left posterior mandible	NR	5	No	S + aRT	96
14	[22]	Iran	27	M	Right posterior mandible	Yes	2	No	S	8 (R)
15	[23]	India	60	F	Left posterior maxilla	No	4	NR	NR	NR
16	[24]	India	42	F	Hard palate	NR	3	NR	NR	NR
17	[25]	Nigeria	16	F	Anterior maxilla	NR	NR	NR	S	30 (R)
18			32	M	Left maxilla	NR	NR	NR	S	NR
19			26	M	Right maxilla	NR	NR	NR	S	12 (R)
20			42	M	Anterior maxilla	NR	NR	NR	S	NR
21			12	F	Posterior maxilla	NR	NR	NR	S	NR
22	[26]	India	57	M	Right tongue	NR	3	No	S	12 (DF)
23	[7]	Japan	8	M	Right posterior mandible	No	2.5	No	nCT + S + aCT	8 (DF)
24	[27]	India	11	M	Left posterior mandible	No	7.9	NR	S + aRT	NR
25	[28]	India	46	F	Right posterior maxilla	Yes	4	NR	S	24 (DF)
26	[29]	Japan	44	F	Right mandible	Yes	3.5	NR	S	48 (DF)
27	[30]	India	14	M	Right mandible	Yes	NR	No	S + aCT + aRT	11 (DOD)
28	[31]	Argentina	61	F	Right mandible	NR	2	Pancreas, stomach	S + aRT	72 (DF)
29	[32]	India	55	F	Hard palate	No	5	Laterocervical lymphnodes	S	12 (DF)
30	[33]	India	55	M	Posterior maxilla	No	NR	NR	CT	8 (DOD)
31			20	M	Palate	No	NR	NR	CT + RT	23 (DOD)
32	[34]	India	32	M	Left mandible	No	NR	No	S	60 (DF)
33	[35]	India	23	M	Left maxilla	Yes	NR	Mediastinum	S + aCT + aRT	3 (DOD)
34	[36]	USA	24	M	Right mandible	NR	6	No	nCT + S + aCT	24 (DF)
35	[34]	Iran	28	M	Right anterior maxilla	Yes	3	Lung	S +aRT	8 (DF)
36	[8]	Brazil	88	F	Right posterior mandible	NR	NR	Lung	S + aCT	12 (DOD)
37	[37]	India	33	M	Left mandible	Yes	5	No	S + aRT	NR
38	[38]	India	36	M	Right posterior mandible	Yes	NR	No	S	24 (DF)
39	[39]	Italy	81	M	Left mandible	No	8	NR	NR	NR
40	[40]	China	45	M	Right maxilla	NR	NR	NR	S	9 (DOD)
41			42	M	Left tongue	NR	NR	NR	S + aRT + aCT	5 (DOD)
42			46	F	Right mandible	NR	NR	NR	nANLO + S + aCT + aRT	18 (DOD)
43			63	F	Left maxilla	NR	NR	Lung	S + aCT	5 (DOD)
44			39	M	Left tongue	NR	NR	NR	S + aCT + aRT	5 (DOD)
45			79	F	Left bucca	NR	NR	NR	S + aCT	6.5 (DOD)
46			49	M	Right tongue	NR	NR	NR	S + aCT + aRT	6 (DOD)
47			20	F	Right palate	NR	NR	Lung	nANLO + S + aCT + aRT	17 (DOD)
48			59	M	Left maxilla	NR	NR	NR	S + aCT + aRT	12 (DOD)
49	[41]	China	49	M	Floor of mouth	NR	5.1	No	S + aCT + aRT	26 (DF)
50	[42]	India	67	M	Left bucca	Yes	6.2	No	S	NR
51	[43]	India	50	F	Floor of mouth	No	4.5	NR	S	NR
52	[44]	USA	69	M	Right cheek	NR	3.2	No	S	6 (DF)
53	[45]	India	68	M	Right posterior mandible	No	5	NR	S	3
54	Present case	Italy	54	F	Left maxilla	No	10	No	nCT + S + aRT	8 (DF)

## Data Availability

The original contributions presented in this study are included in the article. Further inquiries can be directed to the corresponding author.

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
