# Peer review of "Oral Undifferentiated Pleomorphic Sarcoma: A Novel SPECC1L::TERT Gene Fusion and a Comprehensive Literature Review"

_genes, 2025, doi:10.3390/genes16070830_

Round 1

Reviewer 1 Report

Comments and Suggestions for Authors

In this manuscript, the authors present a novel case of oral undifferentiated pleomorphic sarcoma (UPS) and provide a comprehensive review of the literature on oral UPS. While the findings are interesting, the authors need to address the following points:

1. Figure 1, Panel B: For the contrast-enhanced CT scan, please add dotted lines to delineate the heterogeneous soft tissue mass eroding the left maxilla. This will aid in reader comprehension and improve the clarity of the image.

2. Figure 2: Please specify the type of histological staining used in each panel, and include annotations for scale bars in the figure legend.

Panel A: Use dotted lines to demarcate the necrotic and hemorrhagic areas.

Panels B and C: Add arrows and arrowheads to clearly identify the “ectatic, thin-walled vessels” and “scattered multinucleated giant cells,” respectively.

3. Table 1: The “Follow-up” column appears to be only partially visible. Please upload a complete and properly formatted version of the table to ensure the data is fully accessible.

Please include annotations for all abbreviations used in the table, including “NR”, to improve clarity and ensure consistency for readers unfamiliar with the terms.

Author Response

Reviewer n'1: 1. Figure 1, Panel B: For the contrast-enhanced CT scan, please add dotted lines to delineate the heterogeneous soft tissue mass eroding the left maxilla. This will aid in reader comprehension and improve the clarity of the image.

Answer n'1: 1. We have added dotted lines to the CT scan image.

Reviewer n'1: 2. Figure 2: Please specify the type of histological staining used in each panel, and include annotations for scale bars in the figure legend. Panel A: Use dotted lines to demarcate the necrotic and hemorrhagic areas. Panels B and C: Add arrows and arrowheads to clearly identify the “ectatic, thin-walled vessels” and “scattered multinucleated giant cells,” respectively.

Answer n'2: We have included more information regarding the histological features of the lesion. For each image, we have indicated the type of staining, scale bar dimensions, and added dotted lines and arrows/arrowheads to improve the clarity and appreciation of the histological findings.

Reviewer n'1: 

3. Table 1: The “Follow-up” column appears to be only partially visible. Please upload a complete and properly formatted version of the table to ensure the data is fully accessible.

Please include annotations for all abbreviations used in the table, including “NR”, to improve clarity and ensure consistency for readers unfamiliar with the terms.

Answer n'3: We have listed all abbreviations in the table legend. In addition, we have reformatted the table to ensure it displays correctly and completely.

Reviewer 2 Report

Comments and Suggestions for Authors

Thank you for the opportunity to review the manuscript.

The authors present a compelling and well-documented case report of an oral undifferentiated pleomorphic sarcoma (UPS) with a novel SPECC1L::TERT gene fusion. The manuscript is scientifically sound, well-structured, and contributes meaningfully to the understanding of this rare neoplasm, particularly in the context of its molecular characterization.

 After careful evaluation, I recommend with minor revisions.

  1. The discovery of a SPECC1L::TERT gene fusion is novel and important. However, the discussion on the potential oncogenic role of this fusion could benefit from greater depth or speculation supported by any existing literature on related genes or mechanisms.
  2. If any preliminary modeling or inference (e.g., functional domains affected by the fusion) was conducted, even in silico, please briefly mention it. Otherwise, suggest future avenues for functional validation.
  3. I suggest the authors to expand slightly on the clinical reasoning behind the use of neoadjuvant chemotherapy and postoperative radiotherapy. Were there specific risk factors or tumor features that prompted these decisions despite limited evidence in UPS?

Thanks

Author Response

Reviewer n'2

  1. The discovery of a SPECC1L::TERT gene fusion is novel and important. However, the discussion on the potential oncogenic role of this fusion could benefit from greater depth or speculation supported by any existing literature on related genes or mechanisms. 
  2. If any preliminary modeling or inference (e.g., functional domains affected by the fusion) was conducted, even in silico, please briefly mention it. Otherwise, suggest future avenues for functional validation.

Answer n'1-2

1–2. The role of the SPECC1L::TERT fusion transcript remains completely unknown to date. No data regarding this alteration are currently available in the medical literature or in genomic databases (e.g., OncoKB), nor have in silico models been developed. Furthermore, our panel was not suitable for identifying the exact protein domains involved in the fusion. These points have been addressed and clarified in the revised Discussion (lines 315–324).

Reviewer n'2: 3. We have further clarified the rationale behind the use of preoperative chemotherapy and postoperative radiotherapy, both in the Case Description and the Discussion sections (lines 128–129 and 332–336). In details, the use of neoadjuvant treatment was made with the aim to reduce the tumor mass according with the national guidelines for therapy of sarcomas, in order to allow a less demolitive surgery. Adjuvant radiotherapy was adminestered due to the positivity of resection margins.